

# Further Evaluating the Generalized Itô Correction for Accelerating Convergence of Stochastic Parameterizations with Colored Noise

William Johns[1], Lidong Fang[2], Huan Lei[2,3], and Panos Stinis[4]

[1]AI and Data Analytics Division, Pacific Northwest National Laboratory, Richland, Washington, 99354, USA
[2]Department of Computational Mathematics, Science and Engineering, Michigan State University, East Lansing, Michigan, 48824, USA
[3]Department of Statistics and Probability, Michigan State University, East Lansing, Michigan, 48824, USA
[4]Advanced Computing, Mathematics and Data Division, Pacific Northwest National Laboratory, Richland, Washington, 99354

**Correspondence:** William Johns (william.johns@pnnl.gov)

**Abstract.** Stochastic parameterizations are increasingly used in numerical weather prediction to capture statistical properties of unresolved processes and model uncertainties. However, numerical methods developed for deterministic systems may fail to converge to physically meaningful solutions when applied to stochastic systems without modification. A recent study demonstrated the effectiveness of the generalized Itô correction in improving convergence and solution accuracy for a one-dimensional linear test problem with various noise spectra. In this work, we extend the analysis to two nonlinear systems: a modified one-dimensional Korteweg–de Vries equation and a two-dimensional nonlinear shear layer simulation relevant to numerical weather prediction. Both systems are subjected to stochastic advection with varying noise colors and magnitudes. We compare the convergence and solution accuracy of the Itô-corrected scheme to an uncorrected scheme, as well as its computational efficiency relative to a second-order Runge–Kutta method. Our results highlight the effectiveness of the generalized Itô correction in enhancing solution accuracy and convergence while maintaining computational efficiency.

## 1 Introduction

The multi-scale nature of chemical and physical processes in the atmosphere presents significant challenges in numerical simulation. Processes which are not resolved in the temporal or spatial scale but are still important to the time evolution of a model need to be represented with parameterizations Majda et al. (1999). Some unresolved processes cannot be fully described at any instant in time via the resolved processes. Recent studies have focused on addressing this indeterminacy by introducing a stochastic element into parameterizations Berner et al. (2017); Leutbecher et al. (2017). Numerical methods developed for deterministic systems may produce non-physically relevant solutions when naively applied to stochastic systems. The most common discretizations in stochastic analysis are the Itô and Stratonovich interpretation. The Stratonovich interpretation leads to ordinary calculus where the Itô interpretation does not. As climate prediction and weather forecasting rely on fundamentally



continuous processes, which therefore obey the ordinary rules of calculus, the Stratonovich solution is often the more phys­ically relevant interpretation Oksendal (2013). For a thorough discussion of the difference between the Itô and Stratonovich solutions, see chapter 7 in Kloeden and Platen (1992), the review paper by Mannella and McClintock (2012), and Moon and Wettlaufer (2014). Many deterministic numerical schemes converge to the Itô interpretation when applied to stochastic sys­-
25 tems. Throughout this work we will be considering convergence only in the sense of strong convergence, for brevity, we will simply refer to it as convergence. For a detailed discussion of strong convergence and other convergence metrics, see Kloeden and Platen (1992). Fortunately, the Itô and Stratonovich interpretations are related. The convergence of a numerical scheme under the Itô interpretation can be changed to the Stratonovich interpretation with the introduction of a correction term called the *Itô Correction* Oksendal (2013). We note that the addition of the Itô correction can change the order of convergence of
30 the resulting scheme. Although colored noise can be temporally resolved by using small enough time steps to allow the use of deterministic schemes, the required value is often not feasible in practice. A recent work by some of the co-authors intro­-duced a *generalized Itô correction* (GIC) term which is suitable for colored noise Stinis et al. (2020). It was demonstrated that the GIC both improves the final time error and convergence of deterministic numerical schemes with colored noise on a one-dimensional advection-diffusion equation with stochastic forced advection, even for large time steps.

In this work, we demonstrate that the GIC can improve convergence and accuracy on more complicated non-linear systems arising from the numerical weather prediction models. Since analytical solutions for these non-linear systems are not available, we test the convergence of the schemes to a reference solution computed with a very small time step with Heun's second-order Runge-Kutta (RK2) scheme, which converges to the Stratonovich interpretation Hodyss et al. (2013). Additionally, we test that the reference solution has "self converged", that is, taking smaller time steps does not result in significant changes to
the solution. Furthermore, we show that the GIC performs well with increasing magnitudes of colored noise. To demonstrate the flexibility of the GIC we add it to two higher-order schemes and demonstrate its effectiveness at reducing the final error and improving convergence with a 1D homogenous drift-free stochastic differential equation (SDE). We use these examples to highlight how the introduction of the GIC may alter the convergence rate of the scheme. Lastly, we compare the running time of a first-order deterministic numerical scheme (forward Euler) with the GIC to that of the second-order RK2 scheme,
which itself converges to the Stratonovich interpretation. The introdution of the GIC proves to be efficient with negligible computational overhead, providing a scheme with a similar computational cost to the forward Euler (for colored noise) while converging to the Stratonovich solution as desired. For applications where computational cost poses a fundamental constraint and stochastic parameterizations with colored noise are desirable, the addition of the GIC can improve the final error even with large step sizes, and guarantee the convergence to physically relevant solutions as the step size is decreased.



## 2  Models

### 2.1  Time evolution equation and the Generalized Itô Correction

Following the derivation in Stinis et al. (2020) we consider the stochastic differential equation

$$\frac{\partial u}{\partial t}(x,t) = D[u] + P_s[u], \tag{1}$$

where $u = u(x,t)$ is a function of spatial and temporal variables. The term $D[u]$ contains only deterministic terms and all of the stochastic terms are in $P_s[u]$. $\dot{R}(t)$ is a colored noise term which is spatially homogeneous. Without loss of generality, we assume $\mathbb{E}[\dot{R}(t)] = 0$. If $P_s[u]$ has the form

$$P_s[u] = g[u]\dot{R}(t), \tag{2}$$

the GIC at the $j$th time step in differential form is given by $I_j$ with

$$I_j = \frac{1}{2}\frac{\partial(g[u])}{\partial u}g[u]\bigg|_{t_j}\mathbb{E}\big[(\Delta R_j)^2\big], \tag{3}$$

where $t_j$ is the time at the $j$-th time step and $\Delta R_j$ is the $j$-th increment of $R$. The GIC is applied to a numerical integration scheme converging to the Itô solution as follows. At each time step of the numerical integration, $I_j$ is computed and added to the numerical solution of the right-hand-side of Equation (1) for that timestep. The only term in $I_j$ that needs to be computed at each time step is $\frac{\partial(g[u])}{\partial u}g[u]$. We note that in the white noise case, where $\mathbb{E}\big[(\Delta R_j)^2\big] = \Delta t$, the GIC is equal to the Itô correction (see Stinis et al. (2020) for more details).

### 2.2  Noise approximation

We use the following approximation $n(t)$ of the noise process $\dot{R}(t)$ from Hodyss et al. (2013).

$$n(t) = \frac{1}{\sqrt{N_f \Delta t}}\left(C(\omega_0)\frac{b_0}{\sqrt{2}} + \sum_{m=1}^{N_f} C(\omega_m)\big[a_m\sin(\omega_m t) + b_m\cos(\omega_m t)\big]\right),$$

$$C(\omega) = e^{-\alpha\omega^2}, \qquad \omega_m = \frac{2\pi m}{(N-1)\Delta t}, \qquad N_f = (N-1)/2, \tag{4}$$

where $N$ is the number of discrete time levels per unit of time, including the starting and ending time levels. The parameter $\alpha$ controls the color of the Fourier spectrum of $n(t)$ ($\alpha = 0$ corresponds to *white* noise while $\alpha \neq 0$ to *colored* noise). To construct different realizations of the noise process, we sample, for $m = 0, \ldots, N_f$, the coefficients $a_m$ and $b_m$ independently, from the normal distribution $\mathcal{N}(0,1)$ for different initial seeds of a random number generator. It should be noted that $n(t)$ is an *approximate* random noise. The true noise term $\dot{R}(t)$ would contain an infinite number of Fourier modes while $n(t)$ only has a finite number of modes. Nevertheless, in numerical modeling, we can use $n(t)$ to approximate $\dot{R}(t)$.

For the first two cases investigated in this work, the coefficient function $g[u]$ in Equation (2) will take the form $g[u] = \frac{\partial u}{\partial x}$. For this choice of $g[u]$, we have

$$\frac{\partial(g[u])}{\partial u}g[u] = \frac{\partial^2 u}{\partial x^2}. \tag{5}$$



With the true noise process $n(t)$ represented by an infinite number of Fourier modes, the GIC in differential form is given by

$$I_j = \frac{1}{2} \left. \frac{\partial^2 u}{\partial x^2} \right|_{t_j} \lim_{N_f \to \infty} \frac{1}{N_f} \left[ \frac{C(\omega_0)^2}{2} + \sum_{m=1}^{N_f} C(\omega_m)^2 \right]. \tag{6}$$

We note that $I_j \to 0$ as $N \to \infty$ for any colored noise ($\alpha \neq 0$), but not for the white noise ($\alpha = 0$). This is again because

any colored noise can be resolved by taking sufficiently small time steps, at which point the system can be understood as a deterministic one, and the GIC is no longer necessary. As we can only use a finite number of modes numerically as in Equation (4), the GIC in differential form we use in this work is

$$I_j = \frac{1}{2} \left. \frac{\partial^2 u}{\partial x^2} \right|_{t_j} \frac{1}{N_f} \left[ \frac{C(\omega_0)^2}{2} + \sum_{m=1}^{N_f} C(\omega_m)^2 \right]. \tag{7}$$

### 2.3 1D KdV Model

Following Equation 2.7 and the boundary conditions specified in Hodyss and Nathan (2002), we study the following modified Korteweg-de Vries (KdV) equation for the amplitude $A = A(x,t)$ of low frequency atmospheric waves:

$$\frac{\partial A}{\partial t} = -(m_d \frac{\partial^3 A}{\partial x^3} + (m_p(x) + m_n A)\frac{\partial A}{\partial x} + m_g(x)A). \tag{8}$$

The above equation is modified from the traditional KdV equation with the addition of the linear growth term $m_g(x)A$. The dispersion and nonlinear coefficients, $m_d$ and $m_n$, are constant, whereas the linear, long-wave phase speed and growth/decay

coefficients, $m_p(x)$ and $m_g(x)$, are functions of the zonally varying background flow.

We introduce the stochastic perturbation $n(t)$ on the linear advection term of Equation (8) as

$$\frac{\partial A}{\partial t} = -\left[ m_d \frac{\partial^3 A}{\partial x^3} + \left( m_p(x) + m_n A + n(t) \right)\frac{\partial A}{\partial x} + m_g(x)A \right]. \tag{9}$$

Here the coefficient function $g$ in Equation (2) is $g[A] = \partial A/\partial x$ and the GIC correction at each time step $t_j$ is given by

$$I_j^A = \frac{1}{2} \left. \frac{\partial^2 A}{\partial x^2} \right|_{t_j} \frac{1}{N_f} \left[ \frac{C(\omega_0)^2}{2} + \sum_{m=1}^{N_f} C(\omega_m)^2 \right]. \tag{10}$$

### 2.4 2D Model

Following Hodyss et al. (2013), we study a nonrotating, stably stratified, nonhydrostatic Boussinesq fluid bounded above and below (in $z$ direction) by rigid, horizontal boundaries, but periodic in the horizontal ($x$) direction. The governing equations along an $x$-$z$ cross-section may be combined into two equations in two unknowns, the vorticity $\zeta$ and the potential temperature $\theta$, as

$$\frac{\partial \zeta}{\partial t} = -(u\frac{\partial \zeta}{\partial x} + w\frac{\partial \zeta}{\partial z} + \frac{g_0}{\theta_0}\frac{\partial \theta}{\partial x}) + F, \tag{11}$$

$$\frac{\partial \theta}{\partial t} = -(u\frac{\partial \theta}{\partial x} + w\frac{\partial \theta}{\partial z} + w\frac{\partial \theta_0}{\partial z}) + H, \tag{12}$$



along with the relations

$$u = \frac{\partial \psi}{\partial z}, \qquad w = -\frac{\partial \psi}{\partial x}, \qquad \zeta = \nabla^2 \psi, \tag{13}$$

where $\nabla^2$ is the Laplacian operator in the $x$-$z$ plane, and $u$, $w$, $\psi$, $F$, $H$, $g_0$, and $\theta_0$ are the zonal wind, vertical wind, geostrophic pressure (stream function) field, vorticity source, heat source, standard acceleration due to gravity, and reference temperature, respectively (see Appendix D of Hodyss et al. (2013) for more details about the sub-grid parameterizations $F$ and $H$, linear advection, the eddy viscosity, the thermal diffusion, initial conditions, and boundary conditions etc.).

For this study we do not include stochastic perturbations of $F$ or $H$ as in Hodyss et al. (2013) and introduce a stochastic perturbation $n(t)$ on the zonal wind $u$ in Equations (11) and (12) as,

$$\frac{\partial \zeta}{\partial t} = -\left((u+n)\frac{\partial \zeta}{\partial x} + w\frac{\partial \zeta}{\partial z} + \frac{g}{\theta_0}\frac{\partial \theta}{\partial x}\right) + F, \tag{14}$$

$$\frac{\partial \theta}{\partial t} = -\left((u+n)\frac{\partial \theta}{\partial x} + w\frac{\partial \theta}{\partial z} + w\frac{\partial \theta_0}{\partial z}\right) + H. \tag{15}$$

Here, the function $g$ in Equation (2) is $g[\zeta] = \partial\zeta/\partial x$ in Equation (14) and $g[\theta] = \partial\theta/\partial x$ in Equation (15), and the GICs are given by

$$I_j^\zeta = \frac{1}{2}\left.\frac{\partial^2 \zeta}{\partial x^2}\right|_{t_j} \frac{1}{N_f}\left[\frac{C(\omega_0)^2}{2} + \sum_{m=1}^{N_f} C(\omega_m)^2\right], \tag{16}$$

$$I_j^\theta = \frac{1}{2}\left.\frac{\partial^2 \theta}{\partial x^2}\right|_{t_j} \frac{1}{N_f}\left[\frac{C(\omega_0)^2}{2} + \sum_{m=1}^{N_f} C(\omega_m)^2\right] \tag{17}$$

respectively. We make one additional change to the model: where Hodyss et al. (2013) used a $\Delta t$ dependent hyper-diffusion parameter we set this parameter to be constant; this is necessary for convergence analysis.

## 2.5 Model and schemes for testing higher order methods

To demonstrate the effect of the GIC on higher order schemes, we consider, for simplicity, a homogeneous differential equation driven purely by stochastic terms (drift-free) given by

$$\frac{dX(t)}{dt} = X(t)n(t), \tag{18}$$

with initial condition $X(0) = X_0$. For this choice of $g$ and the noise process $n(t)$, the GIC is given by

$$I_j = \frac{1}{2}X\Big|_{t_j} \frac{1}{N_f}\left[\frac{C(\omega_0)^2}{2} + \sum_{m=1}^{N_f} C(\omega_m)^2\right] \tag{19}$$

Numerical results are presented later in the paper for the three time integration schemes summarized below.

The order 1.0 Milstein (MS) scheme for (18) with colored noise can be written as

$$Y_{n+1} = Y_n + Y_n n(t_n)\Delta t + \frac{1}{2}Y_n\left\{(n(t_n)\Delta t)^2 - E[(\Delta R_j)^2)]\right\}, \tag{20}$$





with the approximation $Y_n \approx X(t_n)$. In the case of white noise, where $E[(\Delta R_j)^2)] = \Delta t$, (20) reduces to the traditional MS scheme. This scheme was chosen to demonstrate the effects of adding GIC to a scheme which converges to the Itô solution with order 1.0 where the forward Euler scheme previously used converges with order .5.

Schemes for strong and weak approximation with colored noise are discussed in detail in Milshtein and Tret'yakov (1994). It will sometimes be the case that the order of convergence of a scheme is changed by the addition of the GIC. To demonstrate this, we consider the order 1.5 Strong Taylor Scheme detailed in Kloeden and Platen (1992) section 10.4.1; for brevity we shall call this the KP scheme. We will demonstrate that while this scheme converges to the Itô solution with order 1.5 the addition of the GIC results in a scheme converging to the Stratonovich solution with order 1. The KP scheme for (18) with colored noise can be written as

$$
Y_{n+1} = Y_n + Y_n n(t_n)\Delta t + \frac{1}{2}Y_n \left\{ (n(t_n)\Delta t)^2 - E[(\Delta R_j)^2)] \right\} \tag{21}
$$
$$
+ \frac{1}{2}Y_n \left\{ \frac{1}{3}(n(t_n)\Delta t)^2 - E[(\Delta R_j)^2)] \right\} n(t_n)\Delta t.
$$

We note that for both of these schemes, there is a term equivalent to the GIC with a minus sign. It is therefore more efficient to remove this term and the GIC, rather than redundantly subtracting and adding it.

The third higher order scheme we consider is the Milstein scheme with the highest order derivative approximated with a forward difference as described in Kloeden and Platen (1992) section 11.1.3, we will call this scheme KP2. This scheme was chosen to demonstrate how the GIC cam be applied to multi-step methods. The KP2 scheme for (18) with colored noise can be written as

$$
Y_{n+1} = Y_n + Y_n n(t_n)\Delta t + \frac{1}{2\sqrt{\Delta t}} \left( \tilde{Y}_n - Y_n \right) \left\{ (n(t_n)\Delta t)^2 - E[(\Delta R_j)^2)] \right\} \tag{22}
$$

with supporting value

$$
\tilde{Y}_n = Y_n + Y_n\sqrt{\Delta t}. \tag{23}
$$

When adding the GIC to KP2 we add it only to (22). Unlike the previous two schemes, there is no term equivalent to the subtraction of the GIC, so the GIC must be added explicitly to (22).

## 3   Numerical Results

In this section, we present numerical results demonstrating the effects of including the GIC on the error and rate of convergence of a numerical scheme. For simplicity, we add the generalized Itô correction to the forward Euler scheme which is known to converge to the Itô interpretation with order .5 (Kloeden and Platen, 1992). Our results show that the inclusion of the generalized Itô correction both improves the final error and increases the critical time step for which the scheme begins to converge to the Stratonovich interpretation. As exact solutions for Equations (9), (14), (15) are not available, we compare the forward Euler results, with and without the GIC, to a high fidelity reference solution obtained using Heun's second-order Runge-Kutta (RK2)




scheme, which is known to converge to the Stratonovich interpretation Hodyss et al. (2013). All errors are computed relative to the norm of the reference solution via

$$\epsilon_{\mathrm{m}} = ||u_{\mathrm{m}} - u_{\mathrm{ref}}||/||u_{\mathrm{ref}}|| \tag{24}$$

where $u_{\mathrm{m}}$ denotes the solution computed with a chosen method, $u_{\mathrm{ref}}$ denotes the reference solution computed with RK2 and $||\cdot||$ is the standard $l_2$ norm. To ensure that the RK2 reference solution is sufficiently converged to the true solution, we compute a sequence of RK2 solutions with decreasing time steps and check that the error in Equation (24) becomes very small. For example, Figure 1 shows the convergence of RK2 solutions for Equation (14) to the reference solution obtained with $\Delta t = 10^{-6}$, averaged over 100 realizations of the noise for each color ($\alpha = [0, 10^{-10}, 10^{-8}, 10^{-6}, 10^{-4}, 10^{-2}]$). As the mean error is very small and changes very little for time steps less than $10^{-5}$, we consider the reference solution sufficiently converged to the true solution. The figure will also show the relation between $\alpha$ and the critical time step after which the convergence reaches its maximal theoretical rate (order of convergence). For $alpha$ closer to 0 (closer to white noise) the smaller the critical times step will be. Each realization of the noise is characterized by the two sequences $\{a_m\}$ and $\{b_m\}$ used in its computation. For different values of $\Delta t$, the number of terms $N_f = \frac{N-1}{2}$ changes in Equation (4) since $N$ is determined by $\Delta t$. This provides a consistent sampling of the same noise realization as the time steps are varied.

## 3.1 Convergence by Noise Spectrum

In this experiment, we compare the convergence and final error for forward Euler with and without the GIC for varying colors of the noise. For the 1D KdV model, we use RK2 with $\Delta t_r = 10^{-5}$ for the reference solution. In Figure 2, we integrate for 5 units of time and compare results over 100 realizations of the noise for each of the chosen colors $\alpha = 10^{-3}, 10^{-4}, 10^{-5}, 10^{-6}, 10^{-7}, 10^{-8}, 10^{-9}, 10^{-10}, 10^{-11}, 10^{-12}]$. As expected, for $\alpha = 0$, forward Euler never begins converging as it converges to the Itô interpretation instead of that of Stratonovich. The addition of the generalized Itô correction, in this case, equals the classical Itô correction and results in convergence of order 0.5 in accordance with the theory of Kloeden and Platen (1992). For all non-zero $\alpha$, we see that for sufficiently small $\Delta t$ forward Euler will begin to converge with order 1.0 as the noise is sufficiently resolved in time that the system can be understood as purely deterministic. For non-zero $\alpha$, the addition of the GIC reduces the final error and increases the critical time step after which the scheme achieves the theoretic best convergence rate of 1.0.

For the 2D fluid model Equations (14) and (15), we use RK2 with a time step $\Delta t_r = 10^{-5}$ as the reference solution. We evolve the model deterministically until t=1000 with the initial conditions prescribed in Hodyss et al. (2013) to guarantee that the fluid is sufficiently evolved from the initial condition. In Figure 3, we compare results for forward Euler with and without the GIC over 100 different realizations of the noise for different colors $\alpha = [0, 10^{-8}, 10^{-6}, 10^{-4}, 10^{-2}]$ integrated to t=1001 with time steps ranging from 0.5 to $10^{-5}$. Again $\alpha = 0$ demonstrates the well-known classical Itô correction for white noise and for non-zero $\alpha$ addition of the GIC results in a lower final error and increased maximal time steps for which the scheme achieves the maximal convergence of order 1.0.



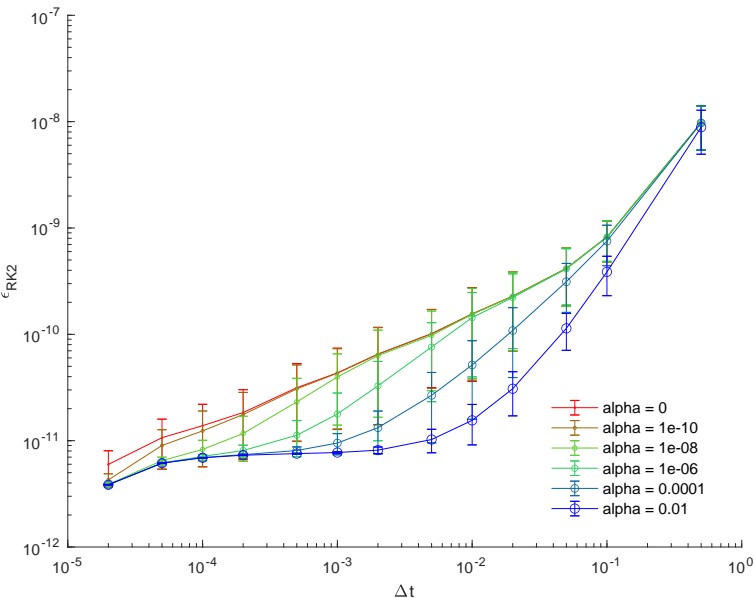

**Figure 1.** Error in the RK2 solution (compared to the reference solution computed with $\Delta t = 10^{-6}$) of the 2D vorticity Equation (14) and the dependency on time step size (horizontal axis) and the characteristics of the noise $\alpha = [0, 10^{-10}, 10^{-8}, 10^{-6}, 10^{-4}, 10^{-2}]$ shown in different colors. Simulations were performed for 100 realizations of the noise process and the solution error was calculated separately for each realization according to Equation (24). The circles are the mean error of the 100 realizations; the vertical bars denote the standard deviation around the mean.

## 3.2 Convergence by Noise Magnitude

In this experiment, we scale the magnitude (equivalently the variance) of the noise with a multiplicative factor $\gamma$ creating a new noise term

$$g^*[u] = \gamma g[u]n(t). \tag{25}$$

Figure 4 shows results averaged over 100 realizations of the colored noise term for the 2D fluid model with $\alpha = 10^{-10}$, $\gamma = [0.01, 0.1, 0.2, 0.5, 1, 2, 5, 10]$ integrated over 1 unit of time with the same initial condition as in Section 3.1. We see that

for all choices of $\gamma$, the critical $\Delta t$ for convergence remains the same $\Delta t_c = 10^{-4}$. Although the critical $\Delta t$ is unchanged, we see that the relative error increases with increasing $\gamma$. This is exactly as expected as the leading error term between RK2 and forward Euler with the GIC is $\mathcal{O}(\gamma^2)$. For brevity, we omit this proof. Although the GIC improves the convergence of forward Euler over the range of magnitudes demonstrated in Figure 4, there is of course a maximal $\gamma$ such that for any larger $\gamma$ the scheme becomes unstable.





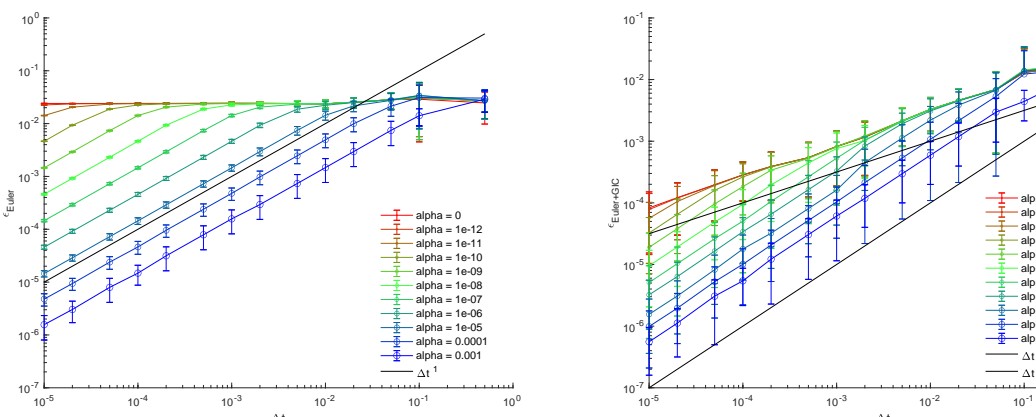

**Figure 2.** Error in the numerical solution (compared to the RK2 reference solution computed at $\Delta t = 10^{-6}$) of the 1D KdV Equation (8) and the dependency on time step size (horizontal axis) and characteristics of the noise term $\alpha = [10^{-3}, 10^{-4}, 10^{-5}, 10^{-6}, 10^{-7}, 10^{-8}, 10^{-9}, 10^{-10}, 10^{-11}, 10^{-12}]$ shown in different colors. Results obtained using the forward Euler scheme without (left) and with (right) the generalized Itô correction, respectively, are shown. Simulations were performed for 100 realizations of the noise process and the relative solution error was calculated separately for each realization using Equation (24). The circles are the mean error of the 100 realizations; the vertical bars denote the standard deviation around the mean. The straight black lines are reference lines indicating convergence rates of 0.5 (upper) and 1.0 (lower), respectively.

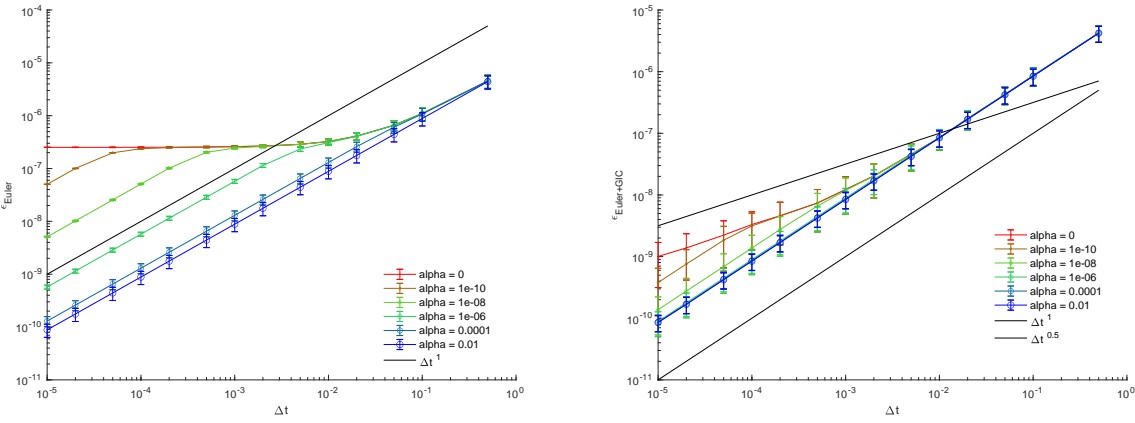

**Figure 3.** Error in the numerical solution (compared to the RK2 reference solution computed with $\Delta t = 10^{-6}$ of the 2D fluid Equation (14) and the dependency on time step size (horizontal axis) and characteristics of the noise term $\alpha = [0, 10^{-8}, 10^{-6}, 10^{-4}, 10^{-2}]$ shown in different colors. Results obtained using the forward Euler scheme (left) without and (right) with the generalized Itô correction, respectively, are shown. Simulations were performed for 100 realizations of the noise process and the relative solution error was calculated separately for each realization using Equation (24). The circles are the mean error of the 100 realizations; the vertical bars denote the standard deviation around the mean. The two lines are reference lines indicating convergence rates of 0.5 (upper) and 1.0 (lower), respectively.





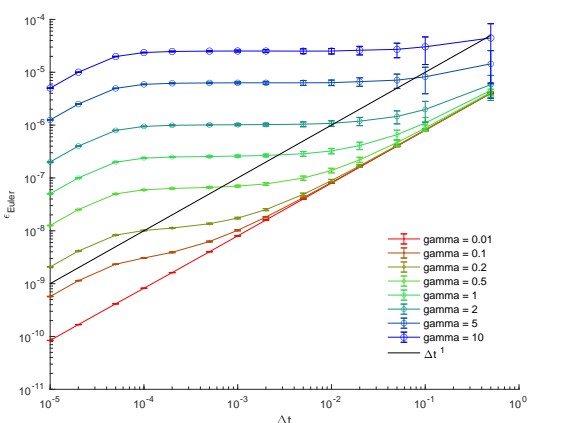 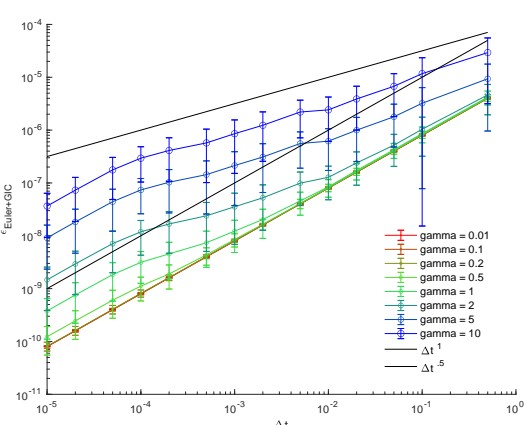

**Figure 4.** Error in the numerical solution of the 2D fluid Equation (14) and the dependency on time step size (horizontal axis) with $\alpha = 10^{-10}$ and scaling factors $\gamma = [0.01, 0.1, 0.2, 0.5, 1, 2, 5, 10]$ shown in different colors. Results obtained using the forward Euler scheme (left) without and (right) with the generalized Itô correction, respectively, are shown. Simulations were performed for 100 realizations of the noise process and the relative solution error was calculated separately for each realization using Equation (24). The circles are the mean error of the 100 realizations; the vertical bars denote the standard deviation around the mean. The black lines are reference lines indicating convergence rates of 0.5 (upper) and 1.0 (lower), respectively.

## 3.3 Adding the GIC to Higher Order Schemes

As forward Euler is a very simple scheme and not a commonly used we next demonstrate how the GIC may be applied to higher order methods. We use the analytic Stratonovich solution to (18) for computing errors in this section in place of the reference solutions used in previous sections.

Figure 5 shows the convergence rate of scheme (20) to the Stratonovich solution with and without the GIC for different colors of noise. Similar to the previous results for forward Euler, the Millstein scheme converges to the Stratonovich solution for all colors of noise and does not converge in the case of white noise, where it converges to the Itô solution. The addition of the GIC decreases the final error for all colors of noise and changes the convergence in the white noise case to the Stratonovich solution. In this case, we see that with the addition of the GIC, the convergence in the white noise case remains order 1.0.

Figure 6 shows the convergence rate of the KP scheme (21) to the Stratonovich solution with and without the GIC for different colors of noise. Again we see that the addition of the GIC reduces the final error and improves the convergence rate for colored noise and changes the convergence of the white noise case to the Stratonovich solution. Note that in the white noise case, the order 1.5 convergence of the KP scheme to the Itô solution has been changed to order 1.0 convergence to the Stratonovich solution. The GIC is an order $\Delta t$ correction term and does not "correct" the higher order terms. Higher order corrections would be necessary to construct an order 1.5 scheme in this manner.

Figure 7 shows the convergence rate of the KP2 scheme (22) to the Stratonovich solution with and without the GIC for different colors of noise. Again we see that the addition of the GIC reduces the final error and improves the convergence rate





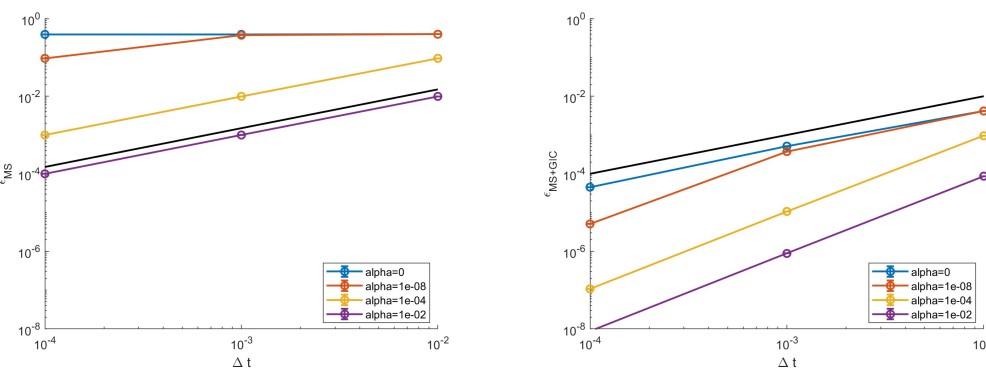

**Figure 5.** Error in the numerical solution of the drift-free SDE (18) and the dependency on time step size (horizontal axis) with $\alpha = [0, 10^{-8}, 10^{-4}, 10^{-2}]$ shown in different colors. Results obtained using the Milstein scheme (left) without and (right) with the generalized Itô correction, respectively, are shown. Simulations were performed for 100 realizations of the noise process and the relative solution error was calculated separately for each realization using Equation (24). The circles are the mean error of the 100 realizations; the vertical bars denote the standard deviation around the mean. The black line is a reference line indicating convergence rate 1.0.

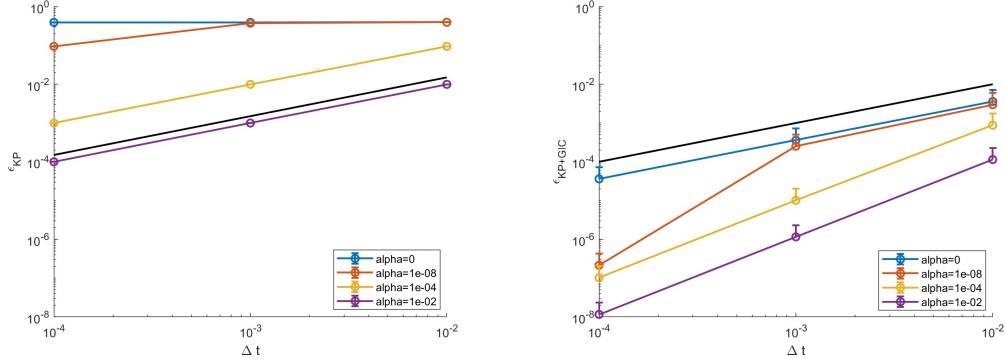

**Figure 6.** Error in the numerical solution of the drift-free SDE (18) and the dependency on time step size (horizontal axis) with $\alpha = [0, 10^{-8}, 10^{-4}, 10^{-2}]$ shown in different colors. Results obtained using KP scheme (left) without and (right) with the generalized Itô correction, respectively, are shown. Simulations were performed for 100 realizations of the noise process and the relative solution error was calculated separately for each realization using Equation (24). The circles are the mean error of the 100 realizations; the vertical bars denote the standard deviation around the mean. The black line is a reference line indicating convergence rate 1.0.

for colored noise and changes the convergence of the white noise case to the Stratonovich solution. We note again that although two separate calculations are performed, one for the supporting value (23) and one for the next time step (23), the GIC is only added to the computation of (23).





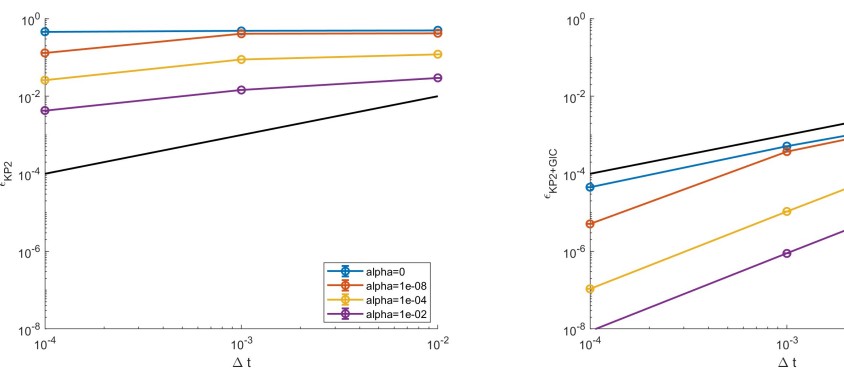

**Figure 7.** Error in the numerical solution of the drift-free SDE (18) and the dependency on time step size (horizontal axis) with $\alpha = [0, 10^{-8}, 10^{-4}, 10^{-2}]$ shown in different colors. Results obtained using KP2 scheme (left) without and (right) with the generalized Itô correction, respectively, are shown. Simulations were performed for 100 realizations of the noise process and the relative solution error was calculated separately for each realization using Equation (24). The circles are the mean error of the 100 realizations; the vertical bars denote the standard deviation around the mean. The black line is a reference line indicating convergence rate 1.0.

## 3.4 Run Time

In this final experiment, we demonstrate the computational efficiency of the GIC. We average the run time of all three schemes considered on the 2D model Equations (14)–(15) over 100 realizations of white noise ($\gamma = 1$) integrated over 50 units of time with $\Delta t = [1, 10^{-1}, 10^{-2}, 10^{-3}]$. Figure 8 shows the ratio of the run time of RK2/Euler and Euler-GIC/Euler. As forward Euler is a first-order scheme (in time) and RK2 is a second-order scheme the ratio of RK2/Euler should be around 2 once enough time steps are taken, this is shown in the orange curve. The ratio of Euler+GIC/Euler remains below 1.1 (blue curve), demonstrating that Euler+GIC remains similar in computational cost. We note that we are solving this model with a pseudo-spectral technique, therefore the additional computation for the GIC at each time step consists of two additional multiplications to compute $\frac{\partial^2 \zeta}{\partial x^2}$ and $\frac{\partial^2 \theta}{\partial x^2}$ in frequency space. In general, the efficiency of computing the GIC for a given model is determined by the difficulty of computing $\frac{\partial g[u]}{\partial u} g[u]$ at each time step compared with the computation of the right hand side of Equation (1).

## 4 Conclusions

This work presents a generalization of the GIC method for non-linear problems from the numerical weather prediction literature, including the modified 1D Korteweg-de Vries (KdV) equation from Hodyss and Nathan (2002) and the 2D (x-y) nonrotating, stably stratified, nonhydrostatic Boussinesq equations from Hodyss et al. (2013). Furthermore, the effect of the GIC is demonstrated for higher-order time integration methods used for solving the 1D drift-free homogeneous stochastic differential equation.



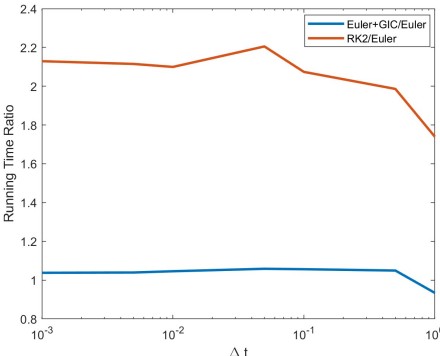

**Figure 8.** Ratio of the running time of RK2/Euler and Euler-GIC/Euler for 2D model Equation (14) averaged over 100 realizations of the noise ($\gamma = 1$) integrated for 50 units of time with $\Delta t = [1, 5 \times 10^{-1}, 10^{-1}, 5 \times 10^{-2}, 10^{-2}, 5 \times 10^{-3}, 10^{-3}]$.

Our numerical experiments demonstrate that when added to a numerical scheme converging to the Itô solution, the GIC alters the convergence of the scheme to the Stratonovich solution, which is the preferred solution for many applications. The GIC proves effective for any color of noise and a large range of magnitudes (or variances) of noise, even in more complex non-linear models. The additional computation of the GIC can be substantially less than using a higher-order scheme and

240 can even be negligible when compared with the run time of the numerical integration of the discretized model. This makes the GIC an attractive option for cheaply computing many potential trajectories of a system for an ensemble and capturing statistical properties about a system. This can be helpful in data assimilation contexts (Van Leeuwen et al., 2019). In addition, the GIC-enhanced solver could be implemented as part of multifidelity approaches (see e.g., Howard et al. (2022) for a recent multifidelity neural operator framework) to provide an efficient low-fidelity estimator. As part of a multifidelity approach, the

245 low-fidelity estimate can then be corrected through the use of *only a few* expensive high fidelity simulations. Finally, while offering computational efficiency, the GIC is also easy to implement and integrate into existing models.

While the model equations used here are still simple compared with the full-fledged weather prediction models, the evaluation presented in this work is a necessary second step following Stinis et al. (2020), which provides the justification and motivation for further exploring the GIC for numerical weather prediction and climate modeling.

*Code availability.* Code for reproducing the experiments and figures in this publication can be found at https://doi.org/10.5281/zenodo.14918193, Johns (2025).

*Author contributions.* This project was concieved by P. Stinis and H. Lei. W. Johns and L. Fang wrote the software, ran the models and preformed the analysis. W. Johns wrote the paper with contributions from all authors



*Acknowledgements.* The authors thank Drs. Christopher J. Vogl (LLNL) and Carol S. Woodward (LLNL) for helpful discussions on the numerical examples shown in this paper and Dr. Daniel Hodyss (NRL) for clarifications regarding his earlier work that inspired our study.



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
