# Peer review of "Further Evaluating the Generalized Itô Correction for Accelerating Convergence of Stochastic Parameterizations with Colored Noise"

_EGUsphere, 2025_

## Referee Comment (RC2)

**Report on**
**"Further Evaluating the Generalized Itô Correction for Accelerating Convergence of Stochastic Parameterizations with Colored Noise"**

The paper proposes and empirically evaluates a generalized Itô correction (GIC) term in numerical schemes to solve PDEs with temporally colored stochastic forcing. The GIC is intended to reduce discretization-induced bias under practical timesteps that may not resolve the temporal scale.

Speccifically, consider numerical integration of the random PDE

$$\partial_t u(x,t) = D[u] + g[u]\dot{R}(t),$$

where $\dot{R}(t)$ is a colored noise process and is approximated by

$$n(t) = \frac{1}{\sqrt{N_f \Delta t}}\left(b_0 + \sum_{m=1}^{N_f} e^{-\alpha\omega_m^2}[a_m \sin(\omega_m t) + b_m \cos(\omega_m t)]\right)$$

with $\omega_m = \frac{2\pi m}{T}$, $N_f = \frac{T}{2\Delta t}$, and $a_m, b_m \sim \mathcal{N}(0,1)$ iid normal random variables. The GIC term has the form

$$I_j = \frac{1}{2}\frac{\partial(g[u])}{\partial u}g[u]\bigg|_{t_j}\mathbb{E}[(\Delta R_j)^2]$$

where $\Delta R = R(t + \Delta t) - R(t)$ is the noise increment over a timestep $\Delta t$. This GIC term resembles the Itô–Stratonovich correction $\frac{1}{2}g(X)g'(X)$ used in SDEs driven by white noise (Recall that Ito SDE

$$dX_t = f(X_t)\,dt + g(X_t)\,dW_t$$

is equivalent to Stratonovich SDE

$$dX_t = [f(X_t) - \frac{1}{2}g(X_t)g'(X_t)]\,dt + g(X_t)\circ dW_t,$$

where the additional drift term is the correction).

This work tests the GIC for non-linear problems from the numerical weather prediction lit- erature, including the modified 1D Korteweg-de Vries (KdV) equation from Hodyss and Nathan (2002) and the 2D nonrotatin nonhydrostatic Boussinesq equations from Hodyss et al. (2013). The numerical experiments demonstrate that when added to a numerical scheme converging to the Itô solution, the GIC alters the convergence of the scheme to the Stratonovich solution, which is the preferred solution for many applications. The GIC proves effective for a large range of colored noises.

**Overall.** The numerical tests are well designed and well-presented, and the numerical convergece demonstrations are convincing in showing the effect of the GIC. I think the paper is a useful contribution to the literature on numerical methods for SDEs with colored noise in numerical weather prediction, and I recommend publication after addressing the following comments.

**Major comments**

- The noise $\dot{R}(t)$ and its approximation $n(t)$ are confusing.
  (1) Why it is the white noise when $\alpha = 0$? Note that when $\alpha = 0$,

$$n(t) = \frac{1}{\sqrt{N_f \Delta t}} \left( b_0 + \sum_{m=1}^{N_f} [a_m \sin(\omega_m t) + b_m \cos(\omega_m t)] \right)$$

  is smooth in $t$ (since $N_f < \infty$) and not a white noise.
  (2) Why does it dependence on $\Delta t$? When $\Delta t = T/N_f$ decreases, $n(t)$ contains more high-frequency components. In other words, the noise becomes "rougher" as $\Delta t$ decreases. This is opposite to the usual understanding of noise approximation, where one takes $n_{t_j}$ of the same noise process at different time resolutions $\{t_j\}$.
  (3) In what sense does the approximation $n(t)$ converge to $\dot{R}(t)$ as $\Delta t \to 0$, partcularly when $\alpha \neq 0$? Clarifying this point would help readers understand the colored-noise model used in this work.
  (4) Why the noice is constant in space? Is this assumption necessary for the GIC to be effective? What if the noise is spatially varying?

- The $\frac{\partial(g[u])}{\partial u} g[u] = u_{xx}$ term in Eq. (5): it should be clarified that $\frac{\partial(g[u])}{\partial u}$ is the Fréchet derivative operator applied to $g[u]$.

- How to compute (6)? In particular, how to approximate $\Delta R$ using $n(t)$?

- Numerical tests: clarify whether the convergence study holds the underlying continuous-time noise path fixed (in an appropriate sense) as $\Delta t$ varies; if not, quantify the effect of the $\Delta t$-dependent noise approximation.